



# Sulphur mass balance and radiative forcing estimation for a moderate volcanic eruption using new sulphate aerosols retrievals based on IASI observations

Henda Guermazi[1,2,3], Pasquale Sellitto[1], Juan Cuesta[1], Maxim Eremenko[1], Mathieu Lachatre[2], Sylvain Mailler[2], Elisa Carboni[4,5], Giuseppe Salerno[6], Tommaso Caltabiano[6], Laurent Menut[2], Mohamed Moncef Serbaji[3], Farhat Rekhiss[3], and Bernard Legras[2]

[1]Laboratoire Inter-universitaire des Systèmes Atmosphériques, UMR CNRS 7583, Université Paris-Est-Créteil, Université de Paris, Institut Pierre Simon Laplace, Créteil, France
[2]Laboratoire de Météorologie Dynamique, UMR CNRS 8539, École Normale Supérieure, PSL research University, École Polytechnique, Sorbonne Universités, École des Ponts PARISTECH, Institut Pierre Simon Laplace, Paris, France
[3]National School of Engineers of Sfax, Water, Energy and Environment Laboratory L3E, University of Sfax, Tunisia
[4]COMET, Atmospheric, Oceanic and Planetary Physics, University of Oxford, Clarendon Laboratory, Oxford, United Kingdom
[5]UK Research and Innovation, Science and Technology Facilities Council, Rutherford Appleton Laboratory, Chilton, United Kingdom
[6]Istituto Nazionale di Geofisica e Vulcanologia, Osservatorio Etneo, Catania, Italy

**Correspondence:** Henda Guermazi(henda.guermazi@lisa.u-pec.fr)

**Abstract.** We developed a new retrieval algorithm based on the Infrared Atmospheric Sounding Interferometer (IASI) observations, called AEROIASI-Sulphates, to measure vertically-resolved sulphate aerosols (SA) extinction and mass concentration profiles, with limited theoretical uncertainties (typically 25% total uncertainty for SA mass column estimations). The algorithm, based on a self-adapting Tikhonov-Phillips regularization method, is applied to a medium-sized-intensity eruption of

Mount Etna volcano (18 March 2012). Comparisons with simultaneous and independent $SO_2$ plume observations and simulations show that AEROIASI-Sulphates correctly identifies the volcanic plume morphology both horizontally and vertically. This method provided, for the first time, crucial information pieces to describe the gaseous-to-particulate volcanic sulphur mass balance (60% of the injected sulphur mass is converted to particulate matter, after 24 hours) and to estimate the regional short-wave direct radiative forcing (a regional forcing of -0.8 W/m$^2$ is exerted in the eastern Mediterranean) for moderate volcanic

eruptions.

## 1 Introduction

Gaining insights into volcanic emissions is important to constrain both internal magma processes and their subsequent atmospheric evolution and impacts. Of particular interest are sulphate aerosols (SA) of volcanic origin that can be both directly emitted (primary SA, (e.g., Roberts et al., 2018)) or produced by the conversion of primary emissions of gaseous sulphur

dioxide ($SO_2$) to particles (secondary SA (e.g., Martin et al., 2014)). The conversion to secondary SA is the major chemical sink of volcanic $SO_2$ emissions and the produced highly-reflective SA are the major source of radiative forcing from volcan-





ism (e.g., Robock, 2000). Moderate to strong explosive eruptions, with stratospheric injection, produce relatively long-lasting perturbations to the hemispheric to global stratospheric aerosol optical depth (AOD) and the Earth's reflectivity in the solar spectral domain (Vernier et al., 2011) and an important radiative imbalance with significant perturbations to the climate system

(Ridley et al., 2014). The impact on the regional radiative budget and climate of smaller tropospheric eruptions and passive degassing emissions is not yet fully understood and estimated, though it was recently suggested that, for certain volcanoes like Mount Etna, it can be significant (Sellitto et al., 2017). Moreover, volcanic SA in the troposphere can degrade air quality (e.g., Michaud et al., 2004), impact ecosystems through acid rain (e.g., Aiuppa et al., 2006) and modify clouds occurrence, lifetime and properties, adding uncertainties to climate modelling (Carslaw et al., 2013).

The quantitative observation of volcanic effluents using satellite instruments is a very powerful tool to get insights into volcanic/atmospheric processes on a systematic spatio-temporal basis. Nevertheless, at present, reliable satellite observations of volcanic effluents are limited to $SO_2$ and, at very specific conditions, other sulphur-containing and halogenated volatiles (Carn et al., 2016, and references therein) and ash (Clarisse and Prata, 2016; Carn and Krotkov, 2016). A partial characterisation of SA can be obtained, for episodic explosive eruptions with relatively high-altitude injections, with satellite observations looking

at the limb (i.e. with a tangent look to the Earth's atmosphere), or using space-based LiDAR (Light Detection And Ranging) instruments (e.g., Vernier et al., 2011). These observations suffer a very scarce spatio-temporal coverage and, in the case of limb observations, a very limited horizontal resolution. This is a fundamental limitation for the process studies at the regional spatial scale. In addition, limb observations are not effective in the troposphere, so limiting the study of tropospheric eruptions and passive degassing activities. Higher spatio-temporal-resolution observations in the nadir geometry (i.e. looking downward

to the Earth) would be much more adapted, in this context. Unfortunately, methods to derive quantitative information on SA from nadir-looking satellite instruments are not available at present, and nadir monitoring is limited to SA qualitative detection at very specific conditions (e.g., Haywood et al., 2010; Karagulian et al., 2010). Despite these limitations, it has recently been shown that high-spectral-resolution nadir infrared instruments, like the Infrared Atmospheric Sounder Interferometer (IASI), have the sufficient information content to provide quantitative information on SA, based on the peculiar SA absorption features

at 1170 $cm^{-1}$ and 905 $cm^{-1}$ (Sellitto and Legras, 2016). In addition, it has been proven that IASI is able to identify and partially characterize given aerosol typologies (Clarisse et al., 2013), based on the known and typology-dependent extinction features provided by its refractive indices. Recently, IASI observations have been used to characterize the vertical profile of coarse aerosol as desert dust and its three-dimensional distribution (Cuesta et al., 2015).

Building on these recent advances, we have developed and we present here a new retrieval algorithm called AEROIASI-

Sulphates, that allows, for the first time, the quantitative observation of the infrared SA extinction profiles and derived mass and optical column properties, using IASI. This algorithm is based on the AEROIASI approach, which is a self-adapting Tikhonov-Phillips regularization method, used previously for dust retrievals (Cuesta et al., 2015), and is described in Sect. 2. AEROIASI-Sulphates is applied to Mount Etna's moderate eruption of 18 March 2012. Results, along with the plume morphological comparison with co-located $SO_2$ retrievals from an independent algorithm and chemistry-transport modelling,

are shown and discussed in Sect. 3. These retrievals are then used to estimate particulate-to-gaseous sulphur mass balance and the regional radiative forcing for this event (Sect. 3). Conclusions are drawn in Sect. 4.





## 2 Data and methods

### 2.1 The IASI instrument

The IASI instrument series are on board MetOp-A -B and -C spacecrafts since 2006, 2012 and 2018, respectively. Each instrument of the series provides a near-global coverage every 12 hours. The IASI is a Fourier transform spectrometer covering the 645-2760 cm$^{-1}$ (3.62-15.5 $\mu$m) spectral range, with an apodized spectral resolution of 0.5 cm$^{-1}$. It offers circular footprints of 12 km radius spaced by 25 km at nadir and a swath of 2200 km (Clerbaux et al., 2009). IASI provides information on meteorological (surface temperature, temperature, humidity profiles and cloud information) and trace gases parameters ($O_3$, CO, $NH_3$ and others). In addition, IASI has been used to observe coarse aerosols like dust (Cuesta et al., 2015; Capelle et al., 2014) and ash (Ventress et al., 2016), and to detect the presence of different kinds of other aerosols (Clarisse et al., 2013).

### 2.2 AEROIASI-Sulphates retrieval scheme

The retrieval scheme presented in this work, hereafter referred to as AEROIASI-Sulphates, follows from the AEROIASI algorithm described by Cuesta et al. (2015). The main setup of the algorithm is summarized in Tab. 1

**Table 1.** AEROIASI-Sulphates main inputs

| Input | Description |
| --- | --- |
| Spectral micro-windows (central wavenumbers in cm$^{-1}$) | 831, 902, 907, 913, 918, 919, 921, 923, 926, 957 |
|  | 975, 979, 1100, 1117, 1155, 1171, 1183, 1204 |
| Mean radius of the SA layer size distribution | 0.52 $\mu$m |
| Standard deviation of the SA layer size distribution | 0.67 |
| SA refractive index | Biermann et al. (2000) |
| $H_2SO_4$ mixing ratio | 57 % |
| A-priori SA profile | Constant and small values of 0.5 particles/cm$^3$ |
|  | in the altitude range 6-21 km |
| R Constraint matrix | Constant with altitude |
| Surface temperature and water vapor profiles, first guess | ERA-Interim reanalyses interpolated for IASI pixels |

The underlying algorithm is a constrained-least-squares fit method based on an auto-adaptive Tikhonov-Philips regularization inverse method (Tikhonov, 1963). The use of a regularization method, rather than most common climatology-based optimal-estimation approaches, is critical in this case because of the fundamental lack of a-priori/climatological information for SA following a given volcanic eruption.




During the iterative solution search, AEROIASI minimises the difference between modelled and measured spectra, through the following cost function, based on Rodgers (2000):

$$\chi^2 = (y - \mathbf{F}(x))^T \mathbf{S}_e^{-1}(y - \mathbf{F}(x)) + (x - x_a)^T \mathbf{R}(x - x_a) \tag{1}$$

In Eq. 1, $\mathbf{S}_e^{-1}$ is the noise error covariance matrix, $y$ and $\mathbf{F}(x)$ are the measured and modeled spectra, $x$ and $x_a$ are the state vector at iteration $i$ and an a-priori state vector and $\mathbf{R}$ is an analytic regularization matrix. The ensemble of $\mathbf{R}$ and $x_a$ in Eq. 1 provides a physical constraint to the otherwise ill-posed inverse problem.

For this study, we use a constant constraint matrix $\mathbf{R}$, where we do not give any altitude-dependent a-priori information.
However, this constraint is adjusted during the iterative procedure following a Levenberg-Marquardt-type method. During the first iterations, the algorithm tries to find the SA layer altitudes and then it modulates the total AODs and the vertical profile shapes in order to minimize the spectral residual. The radiative transfer model $\mathbf{F}$ used in this work is the line-by-line Karlsruhe Optimized and Precise Radiative transfer Algorithm (KOPRA) (Stiller et al., 2002), using as inputs meteorological profiles and first guess of surface temperature and humidity profiles taken from ERA-Interim reanalyses of the European Center Medium
Range Weather Forecast (ECMWF) (Dee et al., 2011). In order to reasonably avoid interferences with water vapour and ozone, we perform the retrieval using 18 micro-windows spectrally narrow from 0.5 to 7 cm$^{-1}$, in the spectral range between 900 and 1220 cm$^{-1}$, characterised by small absorption by these two atmospheric constituents. This spectral range is, at the same time, in a region of strong absorption signature for SA (Sellitto and Legras, 2016). To further limit the interference of water vapour and to provide an adjustable baseline, water vapour profiles and surface temperature are co-retrieved parameters in
AEROIASI-Sulphates. The SA particles are modeled as spherical droplets of a sulphuric acid/water binary solution. The SA refractive indices are taken from Biermann et al. (2000). For this eruption, we assume a 57% sulphuric acid ($H_2SO_4$) mixing ratio, typical for tropospheric volcanic SA and a mono-modal log-normal size distribution with mean radius 0.52 $\mu$m and standard deviation 0.67. The choice of these values are based on the sensitivity analyses performed by Sellitto and Legras (2016).

The output vector is then composed of SA particle number density profiles, water vapour profiles and surface temperature. From SA particle number density, other quantities are derived: SA extinction coefficient and mass concentration profiles, integrated AODs at 10$\mu$m and mass columns.

Quality tests are subsequently applied. AEROIASI-Sulphates retrievals are cloud-screened based on AVHRR (Advanced Very High Resolution Radiometer) data (pixels with high- and medium-altitudes cloud fractions bigger than 20% are screened
out) and differences between the retrieved and the a-priori surface temperatures (pixels with absolute differences bigger than 10K are screened out). Finally, a threshold on Root Mean Square (RMS) spectral residuals is applied (retrievals with RMS lower than 100 nW/(cm$^{-1}$cm$^2$sr are retained).

## 2.3 The eruption of Mount Etna of 18-20 March 2012

On the morning of the 18 March 2012, the New South East Crater (NSEC) gave rise to the 22nd paroxysmal episode of the
long lasting sequence of lava fountains experienced by Mount Etna between 2011 and 2015. Moderate strombolian activity



started on 16 March increasing in intensity on the 18th at about 3:00 (all times here reported are UTC). At about 7:45 the eruptive activity transitioned to discontinuous lava fountain with magma jets became gradually sustained and after 8:00 the height of the lava fountain and eruptive column grew very fast reaching an estimated height of about 7 km above the crater and dispersing at first towards north-east and later east. At about 9:45, the paroxysmal phase started to decline to finish by 10:10, lasting 1:45 hours overall. Moderate strombolian explosions and a weak ash plume continued until 10:25, when the episode eventually stopped.

## 2.4 SO$_2$ plume correlative observations and simulations

SO$_2$ column amounts and plume height for this eruption are retrieved with the optimal-estimation algorithm of Carboni et al. (2012, 2016), using IASI measurements. The algorithm uses the channels between 1000-1200 and 1300-1410 cm$^{-1}$, including two strong SO$_2$ absorption bands. In addition, the same eruption is simulated using the CHIMERE chemistry-transport model (Mailler et al., 2017), for a duration of 120 hours starting on 18 March 2012, 0:00 UTC. In this numerical experiment, SO$_2$ has been treated as a inert tracer: in particular, oxidation of SO$_2$ and subsequent formation of SA have not been represented. Horizontal advection in the CHIMERE model has been represented using the classical Van Leer (1979) second-order slope-limited transport scheme, while the anti-dissipative transport scheme of Després and Lagoutière (1999) has been used for vertical transport. The simulation domain contains 799 x 399 cells at 5 km resolution, with 20 vertical levels from the surface to 150 hPa pressure level. The time and altitude profile for injection of SO$_2$ into the atmosphere has been obtained using the bulk SO$_2$ measured by the ground-based FLAME (FLux Automatic MEasurements) network (Salerno et al., 2018). It is important to notice that this method accurately measure SO$_2$ fluxes during passive degassing and effusive moderate strombolian eruptive activity, in which the plume height is usually maximum of 2 km above the crater. In these cases, an empirical relationship between plume height and wind speed is used to reduce the mass flux rate. Instead, during explosive paroxysmal events where the plume is ejected to higher altitudes, like for this event, this linear height-wind relationship cannot be used and mass flux is retrieved using the plume height estimated by visual camera and/or satellite retrieval.

## 3 Results and discussion

Figure 1a shows the SA plume obtained with AEROIASI-Sulphates, in terms of SA AOD, on 19 March 2012 (IASI daytime overpass, approximate time 9:30 LT). At the overpass time, more than 24 hours after the beginning of the volcanic eruption, the AOD at 10 $\mu$m reaches values as high as 0.2. Two isolated sub-sections of the plume, with higher AOD values, are found: one easterly, in the south of Crete island, and another more westerly, in the south-east direction from Sicily. To gain further insight into the vertical distribution of the SA observed using our algorithm, Fig. 1d shows the altitude of the maximum of the AEROIASI-Sulphates extinction coefficient profile. The two plumes characterized by higher AOD values are located at different mean altitudes. The easterly SA plume (hereafter referred to as *high plume*) has a maximum extinction at about 10-12 km, the westerly SA plume (hereafter referred to as *low plume*) has a maximum extinction at about 6-7 km.



The validation of our SA retrievals is very difficult because of the lack of other SA products from satellite or other adapted ground-based or in-situ datasets. In addition, for this event, CALIOP satellite observations are not available during this day and even this region. Nevertheless, more indirect evaluations of the quality of these retrievals can be performed. In order to

further investigate the spatial consistency of our SA retrievals, they are compared with $SO_2$ observations using the IASI-$SO_2$ Oxford algorithm (Fig. 1b,e) and CHIMERE simulations (Fig. 1c,f). The $SO_2$ and SA plumes, for the same eruptive event and after such a short time from the beginning of the event, are expected to be reasonably co-located in space and time. From Fig. 1, it can be seen that both observations and simulations of the $SO_2$ plume are geographically consistent with AEROIASI-Sulphates retrievals and the two plume sub-sections are generally co-located in space and time, including vertical distributions,

even if IASI-$SO_2$ Oxford *high plume* seems higher than AEROIASI-Sulphates and CHIMERE. This general consistency of the AEROIASI-Sulphates vertical distribution of SA with established (and vertically-resolved) $SO_2$ information underlines the vertical sensitivity of our SA retrievals. The complete time series of modelled $SO_2$ with CHIMERE (data not shown here) reveal that the two plume sub-sections mentioned above splitted during the initial phase of the eruption and were then transported in two different directions. The eastern part of the plume, higher and advected faster towards the east due to the sustained

westerly winds at this altitude can be traced back to the emissions from the paroxystic of the eruption, while the western part of the plume lower and advected more slowly, can be traced back to the parts of the plume emitted either before or after the paroxysmal phase.

The SA mass column concentration is shown in Fig. 2a, for the same overpass. The *high plume* exhibits the largest values in the area, exceeding 1.2 g/m$^2$, while slightly smaller values are found in the *low plume* area. Figure 2b shows the mean SA mass

concentration vertical profile for the *high* and the *low plume*, compared with the mean altitude of IASI-$SO_2$ Oxford retrievals, confirming the vertical separation of the two plumes, which is detected by the two algorithms.

A few areas that cannot be attributed to the volcanic plume are visible in SA retrievals of Figs. 1 and 2, including over Taranto (Italy) and Thessaloniki (Greece) bays. While it cannot be excluded that these large AOD/mass values are false positive detections that are not screened out by our quality control checks, this might also indicate that AEROIASI-Sulphates, even if not

optimised for this task, is sensitive to lower-altitudes anthropogenic sources. These areas are both characterised by particularly low air quality, including a large number of daily exceedences of $SO_2$ thresholds (Kassomenos et al., 2011; Mangia et al., 2013).

The sensitivity and precision of the retrieval has been analysed using the degrees of freedom (DOF, the number of independent parameters that can be theoretically retrieved from the measurements) and total uncertainties (Tab. 2). Due to the

self-adaptation of the inversion algorithm, during the iterative procedure the DOF varies, starting from 1.5 and then reaching typical values of 1.0, during the last iteration. This allows the adaptability of the vertical shape of the profiles, at early iterations, and the convergence of the algorithm, at the end of the procedure. Finally, the average DOF for the *high plume* pixels is 1.04±0.03. These values are an indication that up to two partially independent pieces of information (e.g., the AOD and the altitude of the maximum aerosol extinction) are extracted with AEROIASI-Sulphates, for each IASI observation. Smaller

values are found in a background area, i.e. outside the plume (0.81±0.08). We also provide here an estimation of the total uncertainties on the SA number density profile and mass concentration estimation. This is calculated as the sum of the main


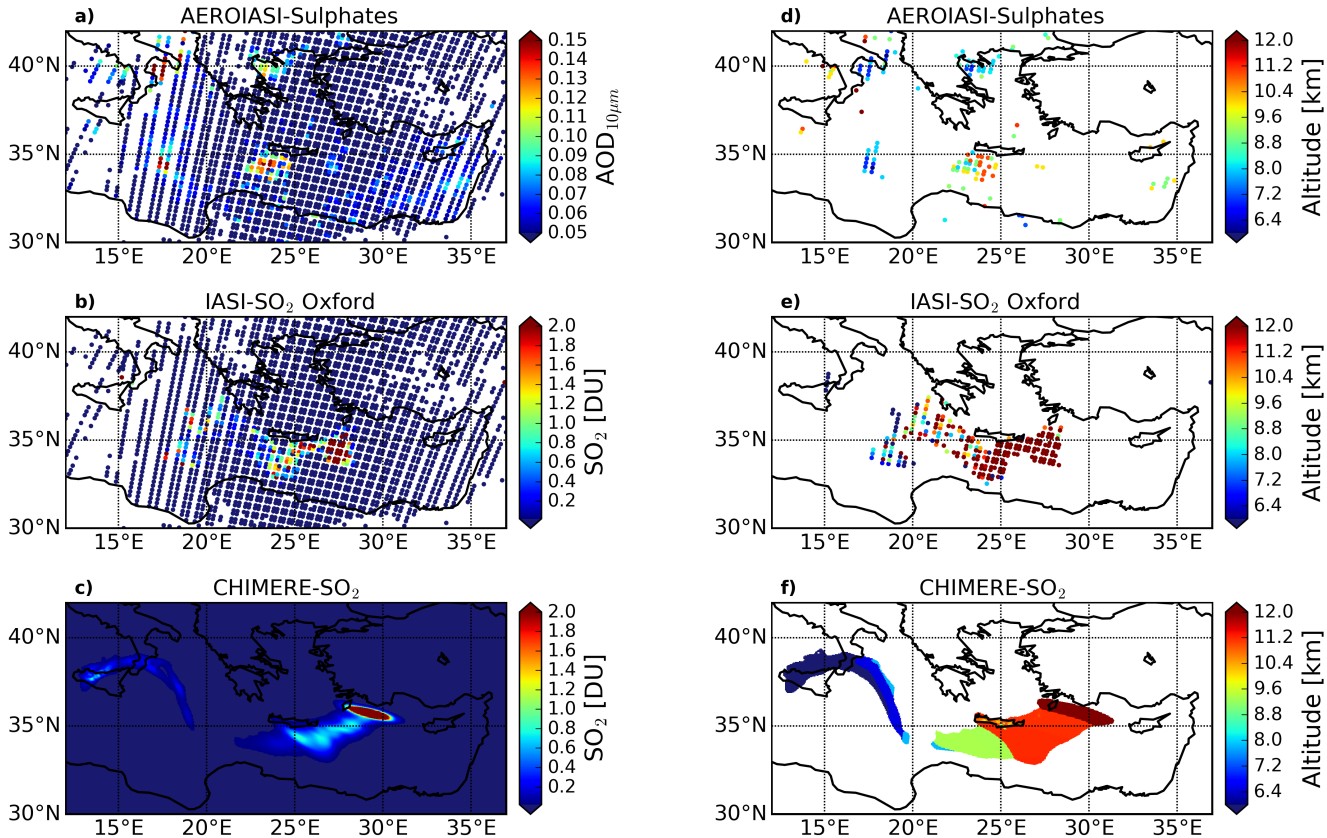

**Figure 1.** AEROIASI-Sulphate AOD (a) and altitude of the aerosol extinction coefficient profile maximum (d); IASI-$SO_2$ Oxford $SO_2$ col-umn amount (b) and altitude of the partial column profile maximum (e); CHIMERE $SO_2$ column amount (c) and altitude of the concentration profile maximum (f), for IASI daytime overpass on 19 March 2012.

random errors (smoothing and the measurement noise error) plus the size-distribution-related systematic error linked to the assumption of a fixed mean radius. Typical uncertainties of 15% are found for the retrieved particle number density profile, that aggregate to obtain a total 24% (15%: random errors, 9%: systematic error on mean radius assumption) on SA mass con-

centration estimations. It must be stressed that systematic error associated with refractive indices (including uncertainties on the sulphuric acid mixing ratio in the SA droplets), and radiative transfer modelling related errors were not taken into account at this stage but can contribute significantly to the error budget.

For this eruption, the total sulphur mass injected in the Mediterranean by Mount Etna is estimated at 1.5 kT, using the FLAME hourly $SO_2$ emission rates integrated during the event. The $SO_2$ retrievals from IASI on 19 March 2012, morning

IASI overpass, allowed the estimation of a gaseous sulphur mass burden of 0.3 kT. Coupling these $SO_2$ mass estimations (total emissions and burden after 24 hours), a $SO_2$ lifetime of about 14h is obtained, which is consistent with previous estimations for

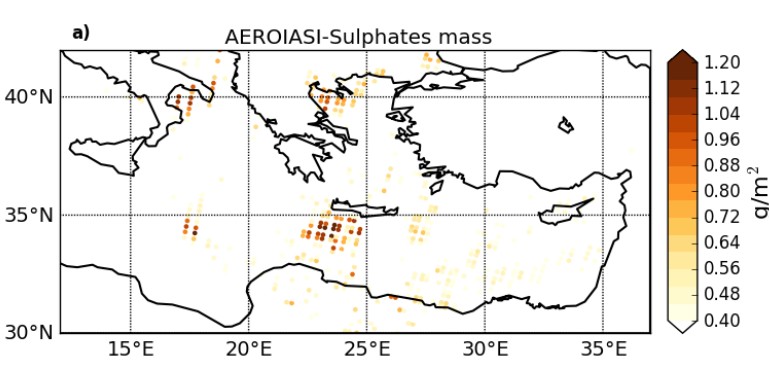
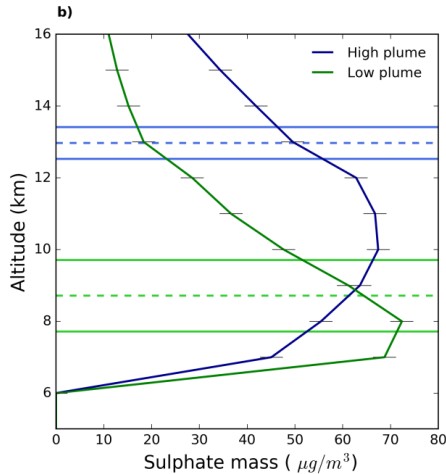

**Figure 2.** AEROIASI-Sulphates mass column concentration for IASI daytime overpass on 19 March 2012 (a). Average verticals profiles of AEROIASI-Sulphates mass concentration for the *high plume* and *low plume* regions. Horizontal lines represent the altitude of the maximum $SO_2$ columns for IASI-$SO_2$ Oxford algorithm.

**Table 2.** Degree of Freedom and uncertainty components for the retrieved SA mass column concentration. In the table, $\epsilon_{sm}$, $\epsilon_m$, $\epsilon_r$, $\epsilon_t$ are the smoothing, measurement noise, radius-related and total errors.

| **DOF** | $\epsilon_{sm} + \epsilon_m$ | $\epsilon_r$ | $\epsilon_t$ |
|---------|------------------------------|--------------|--------------|
| $1.04 \pm 0.03$ | 15 % | 9 % | 24 % |

tropospheric eruptions (e.g., Oppenheimer et al., 1998). Our new SA mass concentration retrievals with AEROIASI-Sulphates allow, for the first time, an estimation of the *missing mass*, converted to particulate sulphur (secondary SA). This mass is estimated at 0.9 kT, corresponding to an instantaneous particulate/gaseous sulphur mass ratio of 3, that can be considered a new

reference ratio for secondary SA formation processes in dispersed volcanic plumes. This estimation contrasts with proximal estimations of this ratio, which have up to two order magnitude smaller values (e.g., Sellitto et al., 2019, and references therein). In that case, SA in the plume should rather be attributed to primary SA emissions. By adding up gaseous and particulate sulphur on 19 March 2012 (a total of 1.2 kT), and comparing to the total injected sulphur, there is still a 0.3 kT missing sulphur mass (20% of the initially injected sulphur mass), that can be probably attributed to the limited sensitivity of satellite observations,

inconsistencies of ground-based and satellite observations or sulphur sinks like particle deposition.

Using the extinction coefficient retrievals from AEROIASI-Sulphates, a rough estimate of the clear-sky shortwave direct radiative effect (DRE) is provided for this event. With a similar method as in Sellitto et al. (e.g., 2016), the observed average SA extinction profile for the *high plume* has been used as input of radiative fluxes calculation by the LibRadtran suite (Mayer and Kylling, 2005), to compute the average radiative effect at the top of atmosphere (TOA) ($DRE_{TOA}$) and at the surface ($DRE_{surf}$)



as well as their f ratio. Even if the aerosol extinction at $10\mu m$ is observed, some hypotheses are needed to fully describe the optical properties of the volcanic aerosol layer. Typical single scattering albedo (0.99) and phase function (Heyney-Greenstein function corresponding to an asymmetry parameter of 0.5) for a not-absorbing/spherical SA layer has been chosen. The extinction at $10\mu m$ has been extended to the shortwave spectral region using an Ångström exponent (AE) of 1.0. Aerosol layers with $AE \geq 1.0$ are usually associated to volcanic-affected air masses (e.g. Sellitto et al., 2017). A wavelength-independent surface

reflectivity of 0.07, typical of sea surfaces, has been selected. A standard mid-latitude atmosphere is considered. We obtained $DRE_{TOA}$=-80W/m$^2$ and $DRE_{surf}$=-83W/m$^2$, corresponding to a near-1 f ratio. Basing on ground-based LiDAR observations, Romano et al. (2018) have measured and simulated DRE for the overpass of an ashy plume from Mount Etna eruption of 3 December 2015 and found typical instantaneous morning $DRE_{TOA}$=-112W/m$^2$ and $DRE_{surf}$=-145W/m$^2$, for an AOD at 532 nm of about 1.0. In our case, smaller AOD at 532 nm were found for the *high plume* (0.5). By scaling our DREs to 1.0

AOD, we obtain very consistent values at surface ($DRE_{surf}$=-156W/m$^2$) and 50% bigger at the TOA ($DRE_{TOA}$=-150W/m$^2$), with respect to Romano et al. (2018). Nevertheless, it has to be noted that for 3 December 2015 the volcanic plume contained probably a substantial fraction of absorbing ash particles, that can explain the smaller forcing at TOA. The presence of ash and the fact that was a particular strong eruption are also responsible for the bigger AOD on this date. The plume's DRE can be averaged over the Eastern Mediterranean basin to obtain an estimate of -0.8W/m$^2$ regional $DRE_{TOA}$ in this area. The Eastern

Mediterranean is the prevalent dispersion sector with 80% of Mount Etna emissions dispersing in this direction (Sellitto et al., 2017). This regional mean DRE is up to 3 times bigger than the hemispheric mean DRE for recent moderate stratospheric eruptions, like for Sarychev (Haywood et al., 2010), even if the impact is on a much smaller area (regional versus hemispheric) and duration (a few days versus several months).

## 4   Conclusions and perspectives

We have presented a new retrieval algorithm (AEROIASI-Sulphates) for the observation of SA using IASI measurements. This inversion scheme, based on a self-adapting Tikhonov-Phillips regularization method, provides, for the first time from space, quantitative vertically-resolved observations of SA extinction and mass concentration with limited uncertainties. AEROIASI-Sulphates algorithm is applied to a medium-sized-intensity eruption of Mount Etna volcano (18 March 2012). This method allows estimating the mass of the converted $SO_2$ emissions to SA (0.9 kT, i.e. 60% of the initially injected sulphur mass, after

24h since the start of this eruptive event), which is important to have a more complete view on the sulphur mass balance for volcanic eruptions. In addition, this method gives access to crucial parameters for the estimation of the DRE. Basing on that, a regional $DRE_{TOA}$ of -0.8 W/m$^2$, in the Eastern Mediterranean sector, has been estimated for this event. This is the first time such radiative forcing estimation is obtained using SA extinctions profiles observations from space. For the future, we aim at the co-retrieval of $SO_2$ and SA, using IASI. It has been recently demonstrated that IASI measurements contain sufficient

information to characterise both species with the same radiance spectra (Guermazi et al., 2017). A synoptic view of gaseous and particulate sulphur emissions, while of great importance to get insights into both inner volcanic and atmospheric processes, is also expected to reduce biases in both retrievals (Sellitto et al., 2019).





*Author contributions.* H.G and P.S conceived the study, analyzed and interpreted the results. H.G. realised the inversions. H.G, J.C and M.E developed the AEROIASI-Sulphates software. E.C realised the SO$_2$ IASI inversion. M.L and S.M performed the chimere simulations. G.S and T.C took SO$_2$ measurements using the FLAME network. All authors discussed the results and contributed to the final manuscript.

*Competing interests.* The authors declare that they have no conflict of interest.

*Acknowledgements.* The authors are grateful for the financial support of: EC 7th Framework Program under grant No. 603557 (StratoClim), CNES under grant TOSCA/IASI, PNTS under grant MIA-SO$_2$, ANR under grant TTL-Xing and AID (Agence de L'Innovation de Défense) under grant TROMPET. We acknowledge the AERIS database and support for providing IASI Level 1 and AVHRR-CLARA2 data, and ECMWF for providing meteorological analyses. CHIMERE simulations were performed using HPC resources from GENCI TGCC under grant no. A0050110274.



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
