# Peer review of "Sulphur mass balance and radiative forcing estimation for a moderate volcanic eruption using new sulphate aerosols retrievals based on IASI observations"

_Atmospheric Measurement Techniques, 2019_

## Referee Comment (RC1) · Anonymous Referee #2 · 23 Oct 2019

The paper introduces a new retrieval scheme for quantifying sulphate aerosol mass and height from IR nadir measurements with IASI. The new retrieval is demonstrated in a case study of a single paroxysmal episode of Mt Etna. The sulphate aerosol mass retrieval is used together with an independent IASI $SO_2$ retrieval to estimate the gas/particle phase partitioning of $SO_2$ 24 h after the eruption. Further, for this plume the shortwave radiative forcing was calculated.

While the title implies that the manuscript mainly focuses on sulphur mass balance and radiative forcing the paper itself is mainly on the retrieval. In my opinion the new quantitative retrieval of sulphate aerosol from IR nadir merits publication in AMT. However, for none of the three aspects the methods and inputs are described sufficiently and the interpretation and discussions of the results are incomprehensible. Further, I think that the motivation that heads towards lower troposphere reaching degassing events does not match with the rather high-troposphere reaching eruption that is presented as a case study.

Since I really like the idea of retrieving sulphate aerosol from IR nadir measurements, I'd like to suggest to revise the manuscript considering the comments below.

**General comments:**

To me it is not clear which altitude range you want to address with the new sulphate aerosol retrieval. From the introduction I had the impression that you will focus on the lower parts of the troposphere (because you speak about volcanic effluents and "passive degassing activities"), but the retrieval and case study rather focus on the upper troposphere/lower stratosphere (6-21 km). Please be more specific and consistent with respect to the altitude range you want to address. For the introduction I'd like to suggest that you focus more on the benefits of the high horizontal resolution that IASI provides.

The main scope of the paper is the introduction of an altitude-resolving retrieval of sulphate aerosol from IASI. However, it is not clear in how far the retrieval agrees with or differs from the AEROIASI retrieval and plenty of characterising information is missing. Please provide more information on the retrieval and work-flow. Did you use any pre-filter to identify SA-containing spectra, e.g. Clarisse et al. 2013, or did you retrieve SA in all spectra and then filtered out clouds? Will other aerosol types (e.g. wild fire, mineral dust) be filtered out, or retrieved as SA? To which altitude range are you sensitive to? What is the vertical resolution/altitude uncertainty? What are the lower AOD and extinction detection limits? Please show kernel functions and averaging

kernel.

Why did you choose one of the strongest in terms of $SO_2$ injection and highest reaching Etna eruptions during the IASI measurements period to demonstrate the IASI sulphate aerosol retrievals can provide global measurement information in the troposphere, especially for effluents and passive degassing events? Either reconsider your motivation or provide an additional example for a low altitude eruption event.

Concerning the methods and the used inputs a lot of information is missing. You are presenting a quantitative sulphate aerosol retrieval from IR nadir measurements, but basic characterisations are missing (please see detailed comments below). The information provided for all three topics of the paper (sulphate aerosol retrieval, Etna case study, direct radiative effect) is not sufficient to allow for reproduction.

Please also add a section on code and data availability for AEROIASI, AEROIASI-Sulphate, KOPRA, LibRadtran and IASI, AVHRR, IASI $SO_2$ retrieval, respectively.

**Specific comments:**

l 2/3: "vertically-resolved" ... "profiles" Remove one, since this is a tautology.

l 4 and throughout the manuscript: "medium-sized": Please specify what you mean by "medium-sized". Medium sized eruption in terms of injection altitude, $SO_2$ mass, ash mass, eruption duration, damage?

l 16: "highly reflective": Sulphate aerosol is highly reflective in UV/VIS, but highly absorbing in mid-IR. Please be more precise here.

l 17: "Moderate to strong": See comment to line 4. Specify and quantify what you mean with moderate to strong. Are these stratosphere reaching eruptions?

l 19: "... important radiative imbalance ... significant ...": Please specify what you mean

by important radiative imbalance and significant perturbations.

l 20: "smaller tropospheric eruptions": Do you mean with smaller only troposphere reaching eruptions? See comments above. Please clarify.

l 26-28: Which altitude range do you mean? "Effluents" I would expect in the boundary layer and lowermost troposphere. Concerning injections this is not true. The whole suite of limb satellites provides information on stratospheric sulphate aerosol and some (e.g. SAGE II, OSIRIS, MIPAS) even extend into the upper troposphere. Further CALIOP provides information on tropospheric sulphate aerosol in the entire troposphere. If you mean only nadir satellite instruments and/or measurements in the (lower) troposphere, please state so here.

l 28: "... partial characterisation ...": Please specify what you mean by partial. I assume, you mean partial characterisation in terms of altitude (stratosphere and upper troposphere)? Or do you mean partial with respect to the horizontal extent? Or something else?

l 29: What is "relatively high altitude"? 5, 10, 15, 20 km?

l 32: Which "process studies"?

l 33: "... limb observations are not effective in the troposphere ...": This is not true for the upper troposphere. Please see comment above and state more precise.

l 54-60; Section 2.1: The IASI instrument is described here, but not the data. Please state which data you used (level 1, level 2?) and where the data is available.

l 62-97; Section 2.2: To me it did not become clear in how far AEROIASI and AEROIASI-Sulphur agree or differ. Please clarify, which are AEROIASI characteristics and what was modified for AEROIASI-sulphates.

l 81 & Table 1: Please provide window boundary wavenumbers instead of window central wavenumbers for reproducibility.

[Figure]

l 81/82 & Table 1: The numbers indicating the spectral range in the text do not agree with the numbers given in Table 1. Please fix.

l 86: Please provide a reference that 57 % sulphuric acid solution is characteristic for "tropospheric volcanic SA". I'd assume it mainly depends on temperature.

l 87-88: Do you really mean the mean radius of the log-normal distribution? I'd rather consider this value the median radius. Please provide the formula you are using for the log-normal distribution.

l 91-92: What is the vertical range and vertical resolution of your output vector? Do you restrict the retrieval to 6-21 km as indicated in Table 1? Why? Don't you aim at effluents and passive degassing events?

l 93: Which quality tests? What do they test? Please clarify.

l 99-106: Is this detailed description of the Etna eruption your own observation? Please state so, or provide references.

l 108-110: In my opinion the introduction and description of the IASI $SO_2$ retrieval should have an own (short) subsection.

l 110-122: The CHIMERE model and simulation data should be in a separate subsection and should provide additional information on the following questions: Which region is covered by the simulation domain? Which external data drives the transport, reanalyses or global model, something else? How long did the eruption last and to which altitudes did it reach? Please show the altitude time series of the $SO_2$ injection.

l 117-122: This part leaves me puzzled. From this I don't understand how you derived the mass flux. Did you use camera or, satellite data, or both for the case study? Please state what you used and provide a reference.

l 126: "higher AOD": enhanced AOD

Figure 1: Ackerman (1997) found that AOD has to exceed 0.01 in order to result in

a signal that exceeds clear air variability (at $11\mu$m). In Fig. 1a) the background AOD is 0.05. Does this mean that the AOD is enhanced over the entire domain? What is your lower detection limit? Which AOD threshold did you choose to show the plume altitude?

l 138-140: For the plume south of Crete I do agree that there is co-location with the $SO_2$ retrieval and the CHIMERE model simulation. However, for the plume at about 34-36N and 17-18E there is not agreement with the model and only for 3 out of 13 IASI footprints there is agreement with the IASI $SO_2$ retrieval. Please clarify. Did you notice that at the locations where your retrieval shows the highest AODs, the $SO_2$ concentration is at the lowest limit? Was there maybe some cloud/aerosol filtering before the $SO_2$ retrieval?

l 145-147: Was the *low plume* before or after the paroxysmal phase? It cannot be both. Please clarify.

l 152-157: It appears likely that the two aerosol plumes close to Taranto and Thessaloniki are related to pollution. Here it is necessary to know from which altitudes the IASI signal comes from (sensitivity kernel from surface to 21 km). See comment on model description.

l 173-180: Is it correct that the FLAME $SO_2$ emission rates count everything from ground to plume top? What is the FLAME estimation error? For which altitude range is the IASI $SO_2$ retrieval valid, $>6$ km? What is the IASI $SO_2$ error? For which region did you estimate the $SO_2$ mass from your retrieval? From the whole domain shown in Fig. 1 or only the two plumes you attributed to the eruption before? Please explain the "instantaneous particulate/gaseous sulphur mass ratio". What is the meaning of it? Doesn't the ratio sulphate aerosol/$SO_2$ depend on time and OH availability? With respect to an $SO_2$ lifetime of 14 h I don't consider particle/gas $SO_2$ ratios measured after 24 h at several 100 km distance comparable to a fresh plume measured directly in the crater.

l 182-185: Please think a bit more about the uncertainties. Maybe 0.3 kT are already within the error bars of FLAME observations and the IASI retrievals? Also, the IASI sulphate retrieval only considers aerosol mass above 6 km. It is unclear for which altitude range the IASI $SO_2$-retrieval accounts for.

l 186-189: Please describe the LibRadtran, the method, and chosen input parameters shortly in the methods section and explain what the f-ratio is for.

l 191: This part is confusing. In the sentence before you state, that you derived the extinction at 10 $\mu$m. But sulphate aerosol has not a single scattering albedo of 0.99 in the IR. It is rather 0.01. Also, at 10 $\mu$m sulphate aerosol is mainly absorbing. If you refer to the UV/VIS range, please state so.

l 193: Having a fixed particle size and width, why don't you use Mie theory to scale from 10 $\mu$m to UV/VIS?

l 195: Which "standard mid-latitude atmosphere" did you use? US-Standard? Please be more specific.

l 196-201: I don't consider the DRE of a sulphate aerosol plume comparable to to the DRE of an ash plume. The difference is already quite obvious, when comparing the surface and TOA values. For the sulphate plume both DREs are very close, whereas for the ash plume there is a factor of 1.3. How did you scale the from sulphate aerosol DRE with AOD of one to a DRE with AOD of 1.0? Did you consider that the scaling factor from 10 $\mu$m to UV/VIS is significantly different for sulphate aerosol and ash?

l 203-206: Which region did you consider for the averaged DRE? Please indicate in Fig 1 and state in $km^2$.

l 206-208: I don't understand the sense of comparing the DRE of a very localised plume 24 h after the eruption with a 30 days old plume averaged over the entire northern hemisphere. Please clarify.

---

## Referee Comment (RC2) · Anonymous Referee #1 · 30 Oct 2019

The manuscript by Guermazi et al., deals with the retrieval of sulphate aerosols from observations of the IASI satellite sounder of the 18 March 2012 Etna eruption. Vertical profiles are retrieved and the results are discussed and compared together with the corresponding SO2 retrievals. Finally an estimate is giving of the direct radiative forcing due to the aerosols.

I am sorry that I cannot be more positive, but even from a cursory read during the technical review, it is/was obvious that the manuscript was not ready for peer review as it misses several crucial elements, that not only make the science non-reproducible,

but that prevent the reader from judging whether the paper is scientifically sound. As further detailed below, the manuscript in its current state, is indeed very far from being publishable. The missing elements need to be added to the manuscript, and after that it needs to be completely re-reviewed.

Major comments:

1. Spectroscopic evidence

Very strong conclusions are drawn from the retrieval results. However, the interpretation of the results is premature, and the authors should first provide evidence that what is retrieved are actually sulfate aerosols. For this I suggest:

- a figure with an IASI observed spectrum (two spectra ideally, one in the high and one in the low altitude part), together with the spectral fit, with and without accounting for sulphates, and which show clearly the sulfate signature in the observed spectra (and with a magnitude that is consistent with the retrieved concentration). This plot should also show that a good fit cannot be obtained when sulfates are not included when both SO2 and volcanic ash is accounted for (see also next point)

- it is completely unclear how SO2, volcanic ash and clouds may interfere with the retrieval, which all absorb in the same spectral range, and whether AEROIASI is able to disentangle the three. Which of those are taken into account? How do clouds, volcanic ash and SO2 affect the retrieval?

- an analysis of the spectra outside the volcanic plume where the sulfate mass is non-zero. Rather than speculating on the origin of these enhancements, an analysis of these spectra needs to be carried out. If we cannot trust the retrieval outside the volcanic plume, why would we trust it inside?

- in addition to this, the "geographically consistency" that the authors claim to see between SO2 and the sulfates is far from convincing. Comparing by hand the individual pixels is in any case not the proper way of doing this. For this I would suggest a

colorcoded plot, where all IASI pixels are shown, and where each pixel is color coded (4 colors) along the lines of:

1. no SO2 or sulfates

2. SO2 alone

3. sulfates alone

4. both SO2 and sulfates detected

In addition, the pixels could have a different symbol depending on whether the pixels contains meteorological clouds or not. Such a figure would allow proper discussion of the plume and the retrieval.

2. Retrieval

As pointed out in detail by the other reviewer, the description of the retrieval is not nearly detailed enough. In addition to the very good questions and comments the other reviewer makes, the following need to be addressed:

- justification of the size distribution. From the text it seems that these are based on a sensitivity study (so what IASI can see), rather than on what can be expected physically. The width of the distribution in particular is bound to have a large impact on the retrieved masses, and needs to be justified carefully

- error budget: the 15% uncertainty is completely unrealistic. It is clear that the main uncertainties are not taken into account (width of the size distribution, spectral interferences, altitude). With respect to altitude: the retrieved profile (a 7 km thick layer in case of the high plume), is not a realistic profile, as the actual plume would be very narrow vertically ($\sim$1 km). Not only does it mean that the retrieved sulphate mass will have an error of 100% at all the other altitudes, it also means that the sulphate mass at the actual height of the plume will be largely underestimated. In addition, having a profile this far from reality, will also mean that large errors are introduced on the total column,

due to the fact that the sensitivity of the sulfate aerosols increases with altitude (in other words, if the retrievals were to be redone with the assumption of a narrow profile that peaks at the altitude where the current vertical profile peaks, the results would be very different).

- why is a vertical profile retrieved when the information content is only 1 degree of freedom on the altitude? Would it be not much better to assume a fixed layer thickness and retrieve the center altitude of this layer? (from what I understand, this is how the SO2 altitude is retrieved)

- what assumptions are made on the spectrally varying surface emissivity?

- as mentioned before: the authors need to explain how clouds, SO2 and volcanic ash are taken into account in the retrieval. In this way also, a value of SO2 will be obtained that be compared with the independently retrieved "Oxford" SO2? Vice verse: what is the impact of the sulfate aerosols on the Oxford SO2?

- the 57% H2SO4 mixing ratio (I guess this refers to the refractive index data that was used) needs to be justified.

- there is a dependence of the refractive index data on temperature in the Biermann data, has this been taken into account? If so, for each altitude layer, a different refractive index need to be used. Please detail and justify.

- why was the ERA-Interim data used for the surface temperature and water vapor profiles? I recommend using the much improved ERA5 dataset. What was the input data for the vertical temperature profiles?

- Please provide details on the R matrix other than "constant"? What was the constant?

- Please provide a plot of the cloud cover for the analyzed scene(s) (possibly to be added to the color-coded plot as requested above)

- Please provide a plot of the DOFS in and outside the plume. Note that the DOF

outside of 0.81 seems extremely high, and again calls into question the reliability of the retrieval.

3. Analysis of the plume before and after the 19 March daytime overpass

Extremely bold statements are made in lines 173-185, which are based on the assumption that the retrieval is correct (and thus on the overly optimistic error budget). What would make this analysis (and in fact the entire paper) convincing is an analysis of the $SO_2$ and sulfate burdens 12 hours before and 12 hours after the 19 March daytime orbit. According to the IASI data displayed on http://sacs.aeronomie.be , the different plumes are readily detectable on those overpasses (and indeed as the authors point out, the plume that is shown is more than 24 hours after the eruption started). It is puzzling that the authors do not show their retrieval on these plumes (especially on the first overpass - what would the justification be not to show it?). Having three data points would allow to estimate $SO_2$ lifetime, and $SO_2$ to sulphate conversion rates, and if consistency is found, would convincingly demonstrate the skill of the retrieval. As it stands, all the interpretation on the sulphur mass balance is speculation at best. In addition, the three data points would enable the authors to make a first realistic estimate of the uncertainty of their retrievals (at least on the random error). Addressing this third point, is in my opinion equally important as the previous two points.

4. general comment on the tone of the manuscript

In general the manuscript is well written, but much interpretation and main conclusions (as stated in the abstract) are stated as facts, and these should be formulated much more carefully, especially when so little (or even contradictory) evidence is provided.

Minor comments:

Line 127: in the south of -> to the south of

Line 143: the correct past tense of split is split, not splitted

Line 190: "Even if", what does this refer to?

Line 197: what does "ashy" mean? mineral ash?

Line 203: the fact that was -> the fact that this was

Line 224: chimere -> CHIMERE
* * *

---

## Author Comment (AC1) · 7 Apr 2020

**Reply to reviewers for the manuscript: "Sulphur mass balance and radiative forcing estimation for a moderate volcanic eruption using new sulphate aerosols retrievals based on IASI observations"**

**Introduction**

The authors would like to warmly thank the two anonymous Referees for the detailed and constructive criticism and comments, that allowed us to redefine the methods and to more clearly present the results of this study. Thanks to their comments, we were able to detect two major problems with the version of our AEROIASI-Sulphates retrieval algorithm discussed in the AMTD manuscript, that we have missed before. First, that version of AEROIASI-Sulphates algorithm was very sensitive to clouds (including low clouds) and very large errors were associated to the presence of clouds in the observed scenes. Second, the selected wavelengths were not optimal in terms of the separation of the SA (sulphate aerosols) and the SO2 information. Correspondingly, we completely redesigned AEROIASI-Sulphates, to take into account these two aspects and more minor aspects suggested by the many Referee's comments. The new version of AEROIASI-Sulphates is discussed in the revised manuscript. In particular, in the new version of AEROIASI-Sulphates (with respect to the previous version):

1) Cloud-screening is more demanding: pixels are screened-out also in case of presence of low clouds and for clouds fraction >0 (>20% in the previous version)
2) 20 spectral micro-windows are selected, 19 in the spectral region 830-970 cm$^{-1}$ (in an absorption band of the undissociated sulphuric acid present in the SA droplets, following the refractive indices of Biermann, 2000) and 1 at 1250 cm$^{-1}$ (in another SA-contained sulphuric acid absorption band). These micro-windows safely avoid the SO2 absorption band in the range 1100-1230 cm$^{-1}$, thus being in a region of maximum sensitivity to SA. This was not the case in the previous version of AEROIASI-Sulphates.

Correspondingly, the new retrievals look very different with respect to the ones linked to the previous version of AEROIASI-Sulphates. A systematic consistency has been found of the detected SA plume and what was called before "the western sub-section of the plume" (located at higher altitude than "the eastern subsection of the plume"). The "eastern sub-section of plume" is not detected with the new AEROIASI-Sulphates version and reasons are mentioned for this in the new version manuscript. The new results are described in the new version manuscript, also considering the many suggestions of the Referees.

Further major changes in the algorithm and presentations of the results have been made and are discussed in our specific replies to Reviewers. We list here a few:

1) The impact of hypotheses on the size distribution is studied more in depth and different cases are tested. This enabled a clearer view on the size-distribution-related systematic errors. These aspects are discussed in the specific replies and the new manuscript version.
2) New correlative CHIMERE simulations are used. These are of better quality than the one used before (see also the dedicated manuscript associated with these simulations that has been very recently submitted to the GMDD journal – Lachatre et al., GMDD 2020, referenced in the text).
3) The algorithm name has been changed to AEROIASI-H2SO4 as it is more appropriate description of the inherent spectroscopy of the algorithm, i.e the new algorithm exploits the absorption band linked to the undissociated sulphuric acid ($H_2SO_4$) present in SA droplets in the spectral range 830-970 cm$^{-1}$.

Correspondingly, our manuscript is significantly longer and has 4 new Figures.

Our specific replies (in blue) to the Referees comments (in black) are given below.

We are very happy of the new version of AEROIASI-Sulphates and the manuscript, that has greatly improved thanks to the Referees comments and suggestions, in our opinion. We hope that the Referees share our enthusiasm.

Sincerely,
The Authors

**Referee #1**

The manuscript by Guermazi et al., deals with the retrieval of sulphate aerosols from observations of the IASI satellite sounder of the 18 March 2012 Etna eruption. Vertical profiles are retrieved and the results are discussed and compared together with the corresponding SO2 retrievals. Finally, an estimate is giving of the direct radiative forcing due to the aerosols.

I am sorry that I cannot be more positive, but even from a cursory read during the technical review, it is/was obvious that the manuscript was not ready for peer review as it misses several crucial elements, that not only make the science non-reproducible, but that prevent the reader from judging whether the paper is scientifically sound. As further detailed below, the manuscript in its current state, is indeed very far from being publishable. The missing elements need to be added to the manuscript, and after that it needs to be completely re-reviewed.

**We wish to thank the Referee #1 for her/his comments and suggestions that greatly help developing a new version of AEROIASI-Sulphates and improving the manuscript. We feel that the new version of the manuscript addresses the many constructive comments. Details below.**

**Major comments:**

**1. Spectroscopic evidence**

C1) Very strong conclusions are drawn from the retrieval results. However, the interpretation of the results is premature, and the authors should first provide evidence that what is retrieved are actually sulfate aerosols. For this I suggest:

- a figure with an IASI observed spectrum (two spectra ideally, one in the high and one in the low altitude part), together with the spectral fit, with and without accounting for Sulphates, and which show clearly the sulfate signature in the observed spectra (and with a magnitude that is consistent with the retrieved concentration). This plot should also show that a good fit cannot be obtained when sulfates are not included when both SO2 and volcanic ash is accounted for (see also next point)

**The analysis of the IASI and AEROIASI-Sulphates fitted spectra was the first motivation for our systematic change of configuration of AEROIASI-Sulphates setup (see our Introduction above). For the new version of AEROIASI-Sulphates and the new version manuscript, a new figure (Fig. 1) and inherent discussion was added to show spectroscopic evidence that SA signatures are present in the IASI spectra for this event and that the new version of AEROIASI-Sulphates fits the spectra accounting for SA. SO2 and ash are discussed in the next point.**

C2) It is completely unclear how SO2, volcanic ash and clouds may interfere with the retrieval, which all absorb in the same spectral range, and whether AEROIASI is able to disentangle the three. Which of those are taken into account? How do clouds, volcanic ash and SO2 affect the retrieval?

**In the new version of AEROIASI-Sulphates, clouds are screened-out even in cases of very small cloud fraction and low clouds (see Introduction). In addition, the SO2 absorption signature, albeit evident in plume-pixels in the range 1100-1230 cm$^{-1}$ (see Fig. 1a in the new version manuscript and corresponding discussion), has no impact in the operational spectral micro-windows for the new version of AEROIASI-Sulphates, that are in an independent spectral range. In addition, we now screen-out pixels with a consistent amount of dust, based on AEROIASI-Dust algorithm (Cuesta et al., 2015). Based on proximal onsite visual observations at Mount Etna (one of the co-authors is the INGV responsible of the operational emission monitoring at Mount Etna) ash was not dominant during this event and its possible presence is expected to be even less important after 24 hours of plume dispersion – at the time of IASI overpass). The spectroscopic evidence discussed above (Referee #1 C1), i.e. the fact that the SA signature is sufficient to fully represent IASI spectra, as for the new Fig. 1, is a further indication that ash was not present in the observed plume or its burden was not enough to affect the IASI spectra significantly.**

C3) An analysis of the spectra outside the volcanic plume where the sulfate mass is non zero. Rather than speculating on the origin of these enhancements, an analysis of these spectra needs to be carried out. If we cannot trust the retrieval outside the volcanic plume, why would we trust it inside?

**New insights into the extra-volcanic detections of AEROIASI- Sulphates are given in the revised manuscript. The AEROIASI-Sulphates map for the overpass of 19/03/2012 is compared with a new CHIMERE simulation accounting for anthropogenic SO2 emissions. Most of the extra-volcanic AEROIASI-Sulphates detections**

are geographically co-located with enhanced-SO2 areas from CHIMERE. Spectra in this area also show SA signatures. While this study is not devoted to the observation and study of anthropogenic SA, we suggest that AEROIASI- Sulphates might be partially sensitive to these sources and lower altitudes (to better exploit this sensitivity, a dedicated development would be necessary, e.g. possibly different a priori, constrain matrix, micro-physical hypotheses on SA and spectral micro-windows selection).

C4) In addition to this, the "geographically consistency" that the authors claim to see between SO2 and the sulfates is far from convincing. Comparing by hand the individual pixels is in any case not the proper way of doing this. For this I would suggest a colorcoded plot, where all IASI pixels are shown, and where each pixel is color coded
(4 colors) along the lines of:
1. no SO2 or sulfates
2. SO2 alone
3. sulfates alone
4. both SO2 and sulfates detected
In addition, the pixels could have a different symbol depending on whether the pixels contains meteorological clouds or not. Such a figure would allow proper discussion of the plume and the retrieval.
We added a figure (Fig. 4), as suggested by the Referee #1. Besides what suggested by the Referee, we also added indications of the presence of dust from AEROIASI-Dust. This new figure is indeed very useful to show the consistency of the SO2 and SA "eastern plume" and areas affected by clouds and dust, and we thank the Referee for the suggestion.

**2. Retrieval**
As pointed out in detail by the other reviewer, the description of the retrieval is not nearly detailed enough. In addition to the very good questions and comments the other reviewer makes, the following need to be addressed:

C5) Justification of the size distribution. From the text it seems that these are based on a sensitivity study (so what IASI can see), rather than on what can be expected physically. The width of the distribution in particular is bound to have a large impact on the retrieved masses, and needs to be justified carefully
In Fig. 1, in addition to what discussed in our reply to C1,2 we show spectroscopic evidence that the particles in the plume are larger (mean radius 0.5 $\mu$m) than what usually expected for aged stratospheric SA plumes (usually around 0.2 $\mu$m). This is basically linked to the expected width of the spectral feature around 905 cm-1 (larger for larger particles) and what is observed by IASI in this region. Reasons for this micro-physical characterisation are discussed in the text.

C6) error budget: the 15% uncertainty is completely unrealistic. It is clear that the main uncertainties are not taken into account (width of the size distribution, spectral interferences, altitude). With respect to altitude: the retrieved profile (a 7 km thick layer in case of the high plume), is not a realistic profile, as the actual plume would be very narrow vertically (⌐1 km). Not only does it mean that the retrieved sulphate mass will have an error of 100% at all the other altitudes, it also means that the sulphate mass at the actual height of the plume will be largely underestimated. In addition, having a profile this far from reality, will also mean that large errors are introduced on the total column, due to the fact that the sensitivity of the sulfate aerosols increases with altitude (in other words, if the retrievals were to be redone with the assumption of a narrow profile that peaks at the altitude where the current vertical profile peaks, the results would be very different).
The possibly dominating systematic errors (due to size distribution assumptions, acidity of the SA, temperature of the layer) are more thoroughly discussed in the revised manuscript. We disagree on the hypothesis of an additional systematic error brought by the "incorrect" retrieved vertical shape – which is linked to the vertical sensitivity of the observing system: this source of uncertainty is already accounted for in the smoothing error and is relatively small in terms of column uncertainties (the uncertainties on the AOD or column mass). A figure (Fig. 3) is added to show the averaging kernels and discuss further the vertical sensitivity issue.

C7) why is a vertical profile retrieved when the information content is only 1 degree of freedom on the altitude? Would it be not much better to assume a fixed layer thickness and retrieve the center altitude of this layer? (from what I understand, this is how the SO2 altitude is retrieved)

**The minimisation of the cost function follows what discussed for AEROIASI-Dust (Cuesta et al., 2015). During the first few iterations, the retrieval has larger DOF (of the order of 1.5 DOF or more) and, in this phase, the algorithm searches for an optimal shape of the extinction profile. Then, in order to obtain numerical stable results, the final DOF is fixed to 1.0 (the constrain matrix is adapted, during these iterations, to converge to 1.0 DOF). Please refer to Cuesta et al. (2015) for further details.**

C8) what assumptions are made on the spectrally varying surface emissivity?

**Surface emissivity is taken from a global monthly IASI-derived climatology (Zhou et al., 2011). This is now mentioned in the revised manuscript.**

C9) as mentioned before: the authors need to explain how clouds, SO2 and volcanic ash are taken into account in the retrieval. In this way also, a value of SO2 will be obtained that be compared with the independently retrieved "Oxford" SO2? Vice versa: what is the impact of the sulfate aerosols on the Oxford SO2?

**Please see our replies to Referee #1 C1,2. SO2 cannot be retrieved with our method, which aims at minimising SO2 impact on the SA retrievals, in the present AEROIASI-Sulphates version. So, these mutual impacts of SA on SO2 retrievals and vice-versa are outside the scope area of the present paper, while future work is ongoing to develop combined SO2/SA retrievals**

C10) the 57% H2SO4 mixing ratio (I guess this refers to the refractive index data that was used) needs to be justified.

**We used refractive indices for SA droplets with a 57% sulphuric acid mixing ratio, following Biermann (2000). This value is smaller than the well-known (and measured) 75% for stratospheric SA plumes and was chosen basing on the hypothesis that in troposphere, with larger humidity than the stratosphere, sulphuric acid neutralises with water with a smaller mass ratio. This is consistent with existing micro-physical parameterisations and discussed in the new version manuscript. It has to be noticed that the available values of mixing ratios, in SA refractive index databases, is limited and we had to cope with this choosing a "reasonable value". The lack of detailed laboratory measurements of SA refractive indices is a well-known issue in the community.**

C11) there is a dependence of the refractive index data on temperature in the Biermann data, has this been taken into account? If so, for each altitude layer, a different refractive index need to be used. Please detail and justify.

**As shown by Sellitto and Legras (2016), the variability of the SA extinction with the temperature (except for very big temperature variability, which is not the case of lower atmosphere and, with no doubts, in the limited vertical range of the retrieved profiles) is negligible.**

C12) why was the ERA-Interim data used for the surface temperature and water vapor profiles? I recommend using the much improved ERA5 dataset. What was the input data for the vertical temperature profiles?

**ERA-Interim data were used for temperature profiles and surface temperature and water vapour profiles first guess. As stated in the text, the surface temperature and the water vapour profiles are co-retrieved, in AEROIASI-Sulphates. We agree that the ERA5 datasets are expected to be of improved quality with respect to ERA-Interim; however, 1) the difference between the two datasets is expected to be limited and 2) the water vapour profiles are retrieved in our scheme. For these reasons, we don't expect that the use of ERA5 versus ERA-Interim dataset would significantly affect the quality of the retrieval. Nevertheless, we mention in the text that we plan to use the more recent ERA5 datasets in future versions of AEROIASI-Sulphates.**

C13) Please provide details on the R matrix other than "constant"? What was the constant?

**More details on the R matrix are given in the text (see also our reply to C7).**

C14) Please provide a plot of the cloud cover for the analyzed scene(s) (possibly to be added to the color-coded plot as requested above)

**We added this information (Fig. 2e) and the new colo-coded map (see reply to C4)).**

C15) Please provide a plot of the DOFS in and outside the plume. Note that the DOF outside of 0.81 seems extremely high, and again calls into question the reliability of the retrieval.
**Due to the underlying idea on the DOF/constrain matrix minimisation method expressed in our reply to C4, we feel that a figure of the DOF would bring little information on the sensitivity of AEROIASI-Sulphates. In any case, we disagree that a DOF of 0.81 is high outside the plume. The DOF, in Tikhonov-Phillips retrieval schemes, depend on many factor, the most important being probably the thermal contrast between the surface and the atmosphere (see e.g. Sellitto et al., The effect of using limited scene-dependent averaging kernels approximations for the implementation of fast observing system simulation experiments targeted on lower tropospheric ozone, Atmos. Meas. Tech., 6, 1869–1881, https://doi.org/10.5194/amt-6-1869-2013, 2013). In addition, having the DOF constrained to converge to 1.0, a value of 0.81 is far from being extremely high.**

**3. Analysis of the plume before and after the 19 March daytime overpass**

C16) Extremely bold statements are made in lines 173-185, which are based on the assumption that the retrieval is correct (and thus on the overly optimistic error budget). What would make this analysis (and in fact the entire paper) convincing is an analysis of the SO2 and sulfate burdens 12 hours before and 12 hours after the 19 March daytime orbit. According to the IASI data displayed on http://sacs.aeronomie.be, the different plumes are readily detectable on those overpasses (and indeed as the authors point out, the plume that is shown is more than 24 hours after the eruption started). It is puzzling that the authors do not show their retrieval on these plumes (especially on the first overpass - what would the justification be not to show it?). Having three data points would allow to estimate SO2 lifetime, and SO2 to sulphate conversion rates, and if consistency is found, would convincingly demonstrate the skill of the retrieval. As it stands, all the interpretation on the sulphur mass balance is speculation at best. In addition, the three data points would enable the authors to make a first realistic estimate of the uncertainty of their retrievals (at least on the random error). Addressing this third point, is in my opinion equally important as the previous two points.
**What is shown in the SACS webpage is the SO2 (and other products) for different satellite sensors. From SO2 emissions, sulphate aerosols form after gas-to-particle processes that require timescales consistent with our 24-hours delay since the eruption. So, we analyze the first reasonable overpass in this regard and discard the one on 18 March where we don't observe Sulphates, either because they are not yet formed or because they are not yet enough to be observable by AEROIASI-Sulphates. Thus, the only reliable information on the initial SO2 emissions is the SO2 rates from ground-based FLAME network and we think that the method we use to calculate the SO2 lifetime and conversion rate is better possible in this specific situation. We also feel that the new spectroscopic evidence brought (Referee #1 comment C1) is much more convincing than what shown in the previous version manuscript.**

**4. General comment on the tone of the manuscript:**
In general, the manuscript is well written, but much interpretation and main conclusions (as stated in the abstract) are stated as facts, and these should be formulated much more carefully, especially when so little (or even contradictory) evidence is provided.
**We agree with the Referee. We rephrased many sentences (especially in the Abstract and Conclusions) to be more cautions while expressing our results.**

**Minor comments:**
Line 127: in the south of -> to the south of
Line 143: the correct past tense of split is split, not splitted
Line 190: "Even if", what does this refer to?
Line 197: what does "ashy" mean? mineral ash?
Line 203: the fact that was -> the fact that this was
Line 224: chimere -> CHIMERE
**Done**

**Reviewer 2**

**Summary**

In my opinion the new quantitative retrieval of sulphate aerosol from IR nadir merits publication in AMT. However, for none of the three aspects the methods and inputs are described sufficiently and the interpretation and discussions of the results are incomprehensible. Further, I think that the motivation that heads towards lower troposphere reaching degassing events does not match with the rather high-troposphere reaching eruption that is presented as a case study. Since I really like the idea of retrieving sulphate aerosol from IR nadir measurements, I'd like to suggest to revise the manuscript considering the comments below.

**We wish to thank the Referee #2, as well, for her/his very detailed comments and many suggestions. That was very helpful to clarify many aspects of the manuscript. We feel that the new version of the manuscript addresses these constructive comments and, thanks to that, has greatly improved.**

**General comments**

C1) To me it is not clear which altitude range you want to address with the new sulphate aerosol retrieval. From the introduction I had the impression that you will focus on the lower parts of the troposphere (because you speak about volcanic effluents and "passive degassing activities"), but the retrieval and case study rather focus on the upper troposphere/lower stratosphere (6-21 km). Please be more specific and consistent with respect to the altitude range you want to address.

C2) For the introduction I'd like to suggest that you focus more on the benefits of the high horizontal resolution that IASI provides.

**The topic of our paper and the target of our new SA product is providing (and testing) capabilities of process study and impacts estimation. To do that, nadir observations are the more adapted tool – for the reasons discussed in the manuscript – and are lacking at present. We find that these motivations are much clearer just simply by the integration of the following specific comments by Referee #2 (C14-18) and a few other modifications that we have implemented. So, in our opinion, the new text clarifies this issue. We would like to stress here that we don't aim at a specific altitude range of volcanic injection (though nadir observations might be a quite unique tool for the observations of lower-tropospheric volcanic activity, not accessible for limb sounders) but, as also pointed at by Referee #1, process/impact studies that are accessible only from high horizontal resolution nadir observations. A small note on the word "effluent": as it can be misleadingly linked to pure volcanic degassing, we have modified it to "volcano-emitted gas and particle species" or "volcanic emissions" throughout the text.**

C3) The main scope of the paper is the introduction of an altitude-resolving retrieval of sulphate aerosol from IASI. However, it is not clear in how far the retrieval agrees with or differs from the AEROIASI retrieval and plenty of characterising information is missing. Please provide more information on the retrieval and work-flow.

**The new version of Sect. 2.2 describes with more details AEROIASI-Sulphates and its work-flow (see also our replies to the many Specific Comments on this subject).**

C4) Did you use any pre-filter to identify SA-containing spectra, e.g. Clarisse et al. 2013, or did you retrieve SA in all spectra and then filtered out clouds? Will other aerosol types (e.g. wild fire, mineral dust) be filtered out, or retrieved as SA? To which altitude range are you sensitive to? What is the vertical resolution/altitude uncertainty? What are the lower AOD and extinction detection limits? Please show kernel functions and averaging kernel.

**We did not use any pre-filter to identify SA-containing spectra, the algorithm retrieves the SA and then pixels where the algorithm 1) does not converge, 2) has a spectral fitting RMS larger that a threshold (smaller threshold in the new AEROIASI-Sulphates than the previous version) and 3) for which clouds are detected with an independent product, are filtered out. In the new version of AEROIASI-Sulphates, we also have implemented a post-filtering based on dust detections of the AEROIASI-Dust algorithm. A new detailed spectroscopic evidence of the presence of SA in the plume is shown in the new version manuscript (Fig. 1, see also reply to Referee #1 C1,2). A further clarification: where SA detections are positive, AEROIASI-Sulphates is expected to fit IASI spectra with limited RMSE only if the SA spectral signature is present. This is due to two concurring factors: 1) In the retrieval, spectra are simulated starting from an**

**aerosol signature based on SA refractive indices. This means that if there are other aerosols with a strong extinction feature in the spectra, that have different composition (wildfire particles, mineral dust, etc), this would generate large RMSE and the pixel would be post-screened out. On top of that, dust-dominated pixels are now detected using the independent AEROIASI-Dust algorithm. 2) AEROIASI-Sulphates operates in selected micro-windows that contain the absorption features of SA and where absorption from other absorbing species (in particular water vapour, ozone and SO2) is absent (ozone, SO2) or very small (water vapour). A AK figure (Fig. 3) is shown and the vertical sensitivity of AEROIASI-Sulphates is discussed.**

C5) Why did you choose one of the strongest in terms of SO2 injection and highest reaching Etna eruptions during the IASI measurements period to demonstrate the IASI sulphate aerosol retrievals can provide global measurement information in the troposphere, especially for effluents and passive degassing events? Either reconsider your motivation or provide an additional example for a low altitude eruption event.
**See our reply to Referee #2 C1,2.**

C6) Concerning the methods and the used inputs a lot of information is missing. You are presenting a quantitative sulphate aerosol retrieval from IR nadir measurements, but basic characterisations are missing (please see detailed comments below). The information provided for all three topics of the paper (sulphate aerosol retrieval, Etna case study, direct radiative effect) is not sufficient to allow for reproduction.
**We developed more the three sections and, in our opinion, the new version manuscript is much clearer (and sensibly longer).**

C7) Please also add a section on code and data availability for AEROIASI, AEROIASI-Sulphate, KOPRA, LibRadtran and IASI, AVHRR, IASI SO2 retrieval, respectively.
**Done.**

**Specific comments**

C8) l2/3: "vertically-resolved" ... "profiles" Remove one, since this is a tautology.
**Done**

C9) l4 and throughout the manuscript: "medium-sized": Please specify what you mean by "medium-sized". Medium sized eruption in terms of injection altitude, SO2 mass, ash mass, eruption duration, damage?
**We specified this throughout the text (see also C11 and 13).**

C10) l16: "highly reflective": Sulphate aerosol is highly reflective in UV/VIS, but highly absorbing in mid-IR. Please be more precise here.
**We slightly modified the sentence to account for this.**

C11) l 17: "Moderate to strong": See comment to line 4. Specify and quantify what you mean with moderate to strong. Are these stratosphere reaching eruptions?
**We modified the sentence by specifying that the magnitude metrics for our discussion here is VEI (Volcanic Explosive Index).**

C12) l19: "... important radiative imbalance ... significant ...": Please specify what you mean by important radiative imbalance and significant perturbations.
**We added in the text the quantification proposed by Ridley et al. (2004)**

C13) l20: "smaller tropospheric eruptions": Do you mean with smaller only troposphere reaching eruptions? See comments above. Please clarify.
**As for comment C11, we refer to VEI and we corrected the text accordingly**

C14) l26-28: Which altitude range do you mean? "Effluents" I would expect in the boundary layer and lowermost troposphere. Concerning injections this is not true. The whole suite of limb satellites provides information on stratospheric sulphate aerosol and some (e.g. SAGE II, OSIRIS, MIPAS) even extend into the

upper troposphere. Further CALIOP provides information on tropospheric sulphate aerosol in the entire troposphere. If you mean only nadir satellite instruments and/or measurements in the (lower) troposphere, please state so here.

**We refer here to observations with a high spatiotemporal sampling and horizontal resolution, that can be used to study processes and estimate impacts. We modified the text to reflect this need and the lack of adapted data from this point of view.**

C15) l28: "... partial characterisation ...": Please specify what you mean by partial. I assume, you mean partial characterisation in terms of altitude (stratosphere and upper troposphere)? Or do you mean partial with respect to the horizontal extent? Or something else?

**We specified that we mean "vertical coverage and horizontal sampling". The limitations of limb sounders and space LiDARs is already discussed at L29-31.**

C16) l29: What is "relatively high altitude"? 5, 10, 15, 20 km?

**We specified "upper-tropospheric—stratospheric".**

C17) l32: Which "process studies"?

**We modified the sentence to "This is a fundamental limitation for the study of plume evolution processes and the estimation of impacts at the regional spatial scale."**

C18) l 33: "... limb observations are not effective in the troposphere ...": This is not true for the upper troposphere. Please see comment above and state more precise.

**We modified this line to "…limb observations are not effective in the mid- to lower-troposphere and are very sensitive to the presence of clouds…"**

C19) l54-60; Section 2.1: The IASI instrument is described here, but not the data. Please state which data you used (level 1, level 2?) and where the data is available.

**The following sentence was added at the end of Sect. 2.1: "For this work, IASI Level 1c data (infrared radiance spectra) are used. These data are distributed by EUMETSAT (European Organisation for the Exploitation of Meteorological Satellites)."**

C20) l62-97; Section 2.2: To me it did not become clear in how far AEROIASI and AEROIASI-Sulphur agree or differ. Please clarify, which are AEROIASI characteristics and what was modified for AEROIASI-Sulphates.

**The whole Sect. 2.2 has been re-organised and re-written in parts, so to highlight the differences between AEROIASI (now called explicitly AEROIASI-Dust) and AEROIASI-Sulphates.**

C21) l81 & Table 1: Please provide window boundary wavenumbers instead of window central wavenumbers for reproducibility.

**Done**

C22) l81/82 & Table 1: The numbers indicating the spectral range in the text do not agree with the numbers given in Table 1. Please fix.

**This part has been modified due to the modification of spectral micro-windows discussed in the Introduction of this Reply.**

C23) l86: Please provide a reference that 57% sulphuric acid solution is characteristic for "tropospheric volcanic SA". I'd assume it mainly depends on temperature.

**It depends on temperature and, most important, humidity. The choice of sulphuric acid mixing ratio is bound by the availability of refractive indices (Biermann, 2000). In the new version of Section 2.2., we justify the choice of 57% mixing ratio.**

C24) l87-88: Do you really mean the mean radius of the log-normal distribution? I'd rather consider this value the median radius. Please provide the formula you are using for the log-normal distribution.

**Yes, we mean the mean radius of the log-normal distribution below:**

$$n(r) = \frac{N_0}{r \ln(\sigma_r)\sqrt{2\pi}} e^{-1/2\left(\frac{\ln\left(\frac{r}{rm}\right)}{\ln(\sigma r)}\right)^2}$$

**Where N0 is the total number concentration (in particles $cm^{-3}$), $r_m$ is the mean radius and $\ln(\sigma_r)$ is the unitless standard deviation of $\ln(r/rm)$.**

C27: l 91-92: What is the vertical range and vertical resolution of your output vector? Do you restrict the retrieval to 6-21 km as indicated in Table 1? Why? Don't you aim at effluents and passive degassing events?
**The retrieval goes in fact from surface to 20 km (corrected in the text), with 1-km resolution.**

C28: l 93: Which quality tests? What do they test? Please clarify.
**Quality tests are explained in the text at L93-97.**

C29: l 99-106: Is this detailed description of the Etna eruption your own observation? Please state so, or provide references.
**This was our observation (a co-author is the Responsible of the monitoring of Mount Etna gas emissions at the Italian National Institute of Geophysics and Volcanology – INGV) but we also added a reference in the text (the specific weekly INGV report describing this event).**

C29: l 108-110: In my opinion the introduction and description of the IASI SO 2 retrieval should have an own (short) subsection.
**Done**

C30: l 110-122: The CHIMERE model and simulation data should be in a separate subsection and should provide additional information on the following questions: Which region is covered by the simulation domain? Which external data drives the transport, reanalyses or global model, something else? How long did the eruption last and to which altitudes did it reach? Please show the altitude time series of the SO 2 injection.
**In the meanwhile, a dedicated paper about CHIMERE modelling for this event has been submitted to the Copernicus journal Geoscientific Model Development (presently under review for Open Discussion in Geoscientific Model Development Discussions). This manuscript contains a detailed description of CHIMERE simulations for this event and is referenced in our manuscript. Nevertheless, a few more pieces of information have been added in the present manuscript, in a dedicated subsection: the meteorological data that force these simulations (WRF) and the input volcanological data (SO2 emission rate and altitude).**

C31: l 117-122: This part leaves me puzzled. From this I don't understand how you derived the mass flux. Did you use camera or, satellite data, or both for the case study? Please state what you used and provide a reference.
**Camera data are used and stated in the text.**

C32: l 126: "higher AOD": enhanced AOD
**Done**

C33: l 138-140: For the plume south of Crete I do agree that there is co-location with the SO 2 retrievals and the CHIMERE model simulation. However, for the plume at about 34-36N and 17-18E there is not agreement with the model and only for 3 out of 13 IASI footprints there is agreement with the IASI SO 2 retrieval. Please clarify. Did you notice that at the locations where your retrieval shows the highest AODs, the SO2 concentrations is at the lowest limit? Was there may be some cloud/aerosol filtering before the SO 2 retrievals?
**The results shown on our new version manuscript are now different with respect to the previous version due to the modification of AEROIASI-Sulphates algorithm set-up. Please note the new figure suggested by Referee #1 (comment C4), that show the mutual positions of the SO2 and SA plumes, and the inherent discussion in the text.**

C34: l 145-147: Was the low plume before or after the paroxysmal phase? It cannot be both.
Please clarify.
**Clarified in the new version manuscript's text**

C35: l 152-157: It appears likely that the two aerosol plumes close to Taranto and Thessaloniki are related to pollution. Here it is necessary to know from which altitudes the IASI signal comes from (sensitivity kernel from surface to 21 km). See comment on model description.
**This is no more applicable due to the new results of the new version of AEROIASI-Sulphates.**

C36: l 173-180: Is it correct that the FLAME SO2 emission rates count everything from ground to plume top? What is the FLAME estimation error? For which altitude range is the IASI SO 2 retrieval valid, >6 km? What is the IASI SO 2 error? For which region did you estimate the SO 2 mass from your retrieval? From the whole domain shown in Fig. 1 or only the two plumes you attributed to the eruption before? Please explain the "instantaneous particulate/gaseous sulphur mass ratio". What is the meaning of it? Doesn't the ratio sulphate aerosol/SO 2 depend on time and OH availability? With respect to an SO2 lifetime of 14 h I don't consider particle/gas SO 2 ratios measured after 24 h at several 100 km distance comparable to a fresh plume measured directly in the crater.
**FLAME and IASI SO2 observations are thoroughly described in the references cited in the text. We feel that re-discussing in detail in our manuscript would be just a repetition of existing literature. Nevertheless, if the Referee #2 still thinks that it is useful, we can add more information. The area used to estimate SO2 mass from IASI is what has been identified as "volcanic" and further details are given in the text. For the particle/gas Sulphur ratio, we just consider the remote plume in this case and compare with a similar estimation at more proximal location just as a process reference (primary emissions versus secondary formation.**

C37: C6l 182-185: Please think a bit more about the uncertainties. Maybe 0.3 kT are already within the error bars of FLAME observations and the IASI retrievals? Also, the IASI sulphate retrieval only considers aerosol mass above 6 km. It is unclear for which altitude range the IASI SO 2 -retrieval accounts for.
**IASI SO2 retrievals and AEROIASI-Sulphates retrievals account for all altitudes from surface to 20 km.**

C38: l 186-189: Please describe the LibRadtran, the method, and chosen input parameters shortly in the methods section and explain what the f-ratio is for.
**We clarify these aspects in the text of the new version manuscript (but please consider that many of the input set-up of the radiative simulations were already discussed in the previous version manuscript).**

C39: l 191: This part is confusing. In the sentence before you state, that you derived the extinction at 10 μm. But sulphate aerosol has not a single scattering albedo of 0.99 in the IR. It is rather 0.01. Also, at 10 μm sulphate aerosol is mainly absorbing. If you refer to the UV/VIS range, please state so.
**We refer to the UV/VIS and we clarified it in the text.**

C40: l 193: Having a fixed particle size and width, why don't you use Mie theory to scale from 10 μm to UV/VIS?
**We use the Angström formula, which is linked to the Mie theory.**

C41: l 195: Which "standard mid-latitude atmosphere" did you use? US-Standard? Please be more specific.
**We specified in the revised text.**

C42: l 196-201: I don't consider the DRE of a sulphate aerosol plume comparable to to the DRE of an ash plume. The difference is already quite obvious, when comparing the surface and TOA values. For the sulphate plume both DREs are very close, whereas for the ash plume there is a factor of 1.3. How did you scale the from sulphate aerosol DRE with AOD of one to a DRE with AOD of 1.0? Did you consider that the scaling factor from 10 μm to UV/VIS is significantly different for sulphate aerosol and ash?
**The text has been extended to clarify all these aspects.**

C43: l 206-208: I don't understand the sense of comparing the DRE of a very localised plume 24 h after the eruption with a 30 days old plume averaged over the entire northern hemisphere.
**That was just a reference to compare DRE figures based on what available in the literature. We still think this is a useful reference.**

---

## Author Comment (AC2) · 7 Apr 2020

Dear Referee,

Please find a detailed point-to-point reply to your comments in the Supplement file.

Thank you for your constructive comments and suggestions, that helped improving our manuscript.

Henda Guermazi on behalf of all co-author

[Figure]

Please also note the supplement to this comment:
https://www.atmos-meas-tech-discuss.net/amt-2019-341/amt-2019-341-AC2-
supplement.pdf